# Structural basis for human sterol isomerase in cholesterol biosynthesis and multidrug recognition

Tao Long [1], Abdirahman Hassan [1], Bonne M Thompson[2], Jeffrey G McDonald[1,2], Jiawei Wang[3] & Xiaochun Li [1,4]

3-β-hydroxysteroid-$\Delta^8$, $\Delta^7$-isomerase, known as Emopamil-Binding Protein (EBP), is an endoplasmic reticulum membrane protein involved in cholesterol biosynthesis, autophagy, oligodendrocyte formation. The mutation on EBP can cause Conradi-Hunermann syndrome, an inborn error. Interestingly, EBP binds an abundance of structurally diverse pharmacologically active compounds, causing drug resistance. Here, we report two crystal structures of human EBP, one in complex with the anti-breast cancer drug tamoxifen and the other in complex with the cholesterol biosynthesis inhibitor U18666A. EBP adopts an unreported fold involving five transmembrane-helices (TMs) that creates a membrane cavity presenting a pharmacological binding site that accommodates multiple different ligands. The compounds exploit their positively-charged amine group to mimic the carbocationic sterol intermediate. Mutagenesis studies on specific residues abolish the isomerase activity and decrease the multidrug binding capacity. This work reveals the catalytic mechanism of EBP-mediated isomerization in cholesterol biosynthesis and how this protein may act as a multi-drug binder.

[1] Department of Molecular Genetics, University of Texas Southwestern Medical Center, Dallas, TX 75390, USA. [2] Center for Human Nutrition, University of Texas Southwestern Medical Center, Dallas, TX 75390, USA. [3] State Key Laboratory of Membrane Biology, School of Life Sciences, Tsinghua University, Beijing 100084, China. [4] Department of Biophysics, University of Texas Southwestern Medical Center, Dallas, TX 75390, USA. Correspondence and requests for materials should be addressed to X.L. (email: xiaochun.li@utsouthwestern.edu)

Cholesterol maintains membrane structure, is a precursor in the biosynthesis of steroid hormones and bile acid, and has various roles in cell signaling. Acetyl-CoA, the precursor of cholesterol biosynthesis, is converted into lanosterol, the first sterol-like intermediate, through a series of reactions in the endoplasmic reticulum[1–3]. Due to its low solubility, lanosterol is handled by a number of membrane enzymes and is eventually converted to cholesterol[4–6] (Supplementary Fig. 1). To date, among those membrane enzymes only the structure of sterol C14-demethylaase[7] (CYP51) and the homolog of $\Delta^{14}$-sterol reductase[4] have been determined. EBP catalyzes the conversion of $\Delta^8$-sterols (e.g zymosterol and zymostenol) to their corresponding $\Delta^7$-isomers (Fig. 1a). Mutations in EBP can lead to Conradi–Hunermann syndrome, which commonly causes growth deficiency, short stature, and curvature of the spine[8].

Inhibition of EBP causes an accumulation of its substrates zymosterol and zymostenol, contributing to autophagy in tumor cells[9,10] and oligodendrocyte formation in the central nervous system[11]. Notably, EBP binds an abundance of structurally diverse pharmacologically active compounds, including antidepressants, antipsychotics, opioid analgesics, sterol biosynthesis inhibitors and anti-tumor reagents[12–14] (Supplementary Fig. 2). This type of broad specificity is similar to the σ1 receptor that has been linked to a wide variety of signal transduction pathways[15], although the sequence analysis shows that EBP and σ1 receptor share no structural similarity. Remarkably, as a component of the microsomal anti-estrogen-binding site (AEBS), which is involved in estrogen receptor-independent effects of tamoxifen, EBP can lower the availability of intracellular tamoxifen, causing resistance[16]. Some EBP ligands have been shown to cause the death of cancer cells by influencing cholesterol metabolism[17,18].

Bioinformatics analysis shows that EBP shares structural features with both the membrane protein TM6SF2, which is associated with nonalcoholic fatty liver disease[19], and the σ2 receptor, which is highly expressed in multiple types of cancer cells[20,21]. Here we present two structures of human EBP protein each in complex with a different pharmacologically active compound, revealing its mechanism of action in cholesterol biosynthesis and multidrug recognition.

## Results

**Functional characterization**. To validate the function of EBP, the human EBP-encoding plasmid was transferred to a yeast sterol isomerase *erg2* knockout strain[22] (Fig. 1b). The expression of human EBP, but not the vector alone, allowed the yeast to survive under exposure to 50 ng/ml cycloheximide, suggesting that human EBP functions as a sterol isomerase in this system (Fig. 1b). However, when we supplemented the medium with either U18666A (an inhibitor of cholesterol biosynthesis and Niemann-Pick C1 protein)[23,24] or tamoxifen (Fig. 1c), growth of the yeast was inhibited (Fig. 1b). Our competition binding assay shows that either U18666A or tamoxifen can compete with the [3H]-Ifenprodil binding of purified EBP in vitro[12] (Fig. 1d). This observation is consistent with a previous ligand-binding study in the yeast microsome[12], suggesting that these compounds may bind the catalytic site of EBP to block enzymatic activity.

**The overall structure**. The purified EBP protein presented a monodisperse peak on gel filtration encouraging us to continue

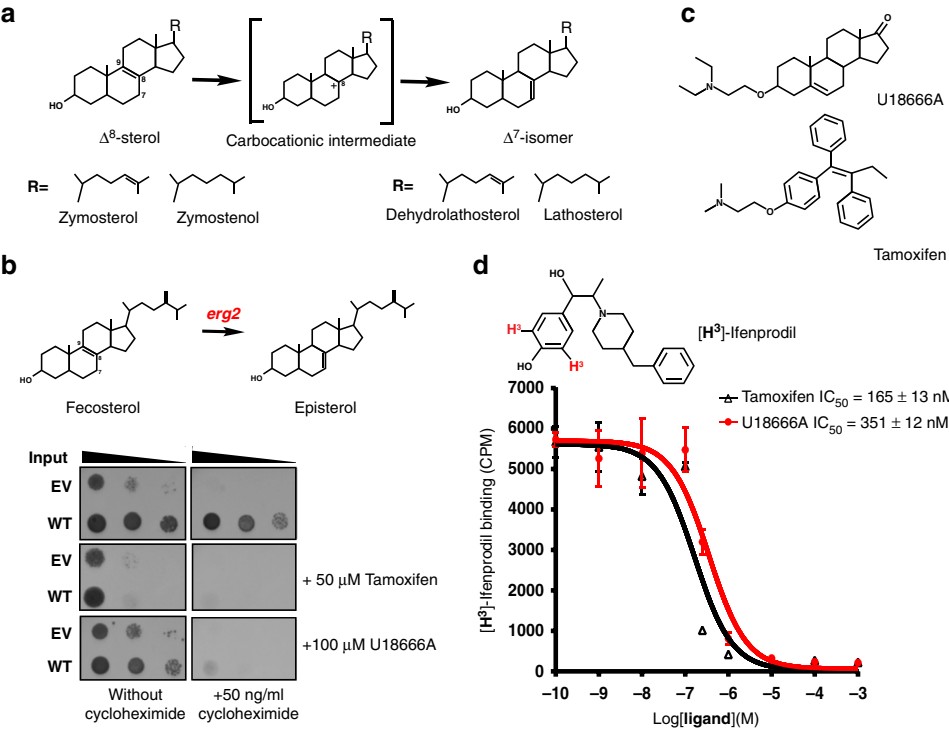

**Fig. 1** Functional characterization of human EBP protein. **a** Reaction catalyzed by human EBP in which zymosterol ($\Delta^8$-sterol) is converted to dehydrolathosterol ($\Delta^7$-sterol). **b** Yeast complementation assay. EBP can rescue the growth of a *Saccharomyces cerevisiae* sterol isomerase Erg2 (yeast EBP homologue) deletion strain (ΔErg2). Growth of yeast expressing human EBP in the presence of sub-inhibitory concentrations of cycloheximide for 24 to 48 h with or without pharmacological compound. **c** The structures of U18666A and tamoxifen. **d** The binding of EBP to different ligands. Inhibition of [3H]-Ifenprodil binding to the purified EBP protein by U18666A (red) and tamoxifen (black). Data shown are the mean ± SD of three determinations. Source data are provided as a Source Data file. EV empty vector, WT wild type

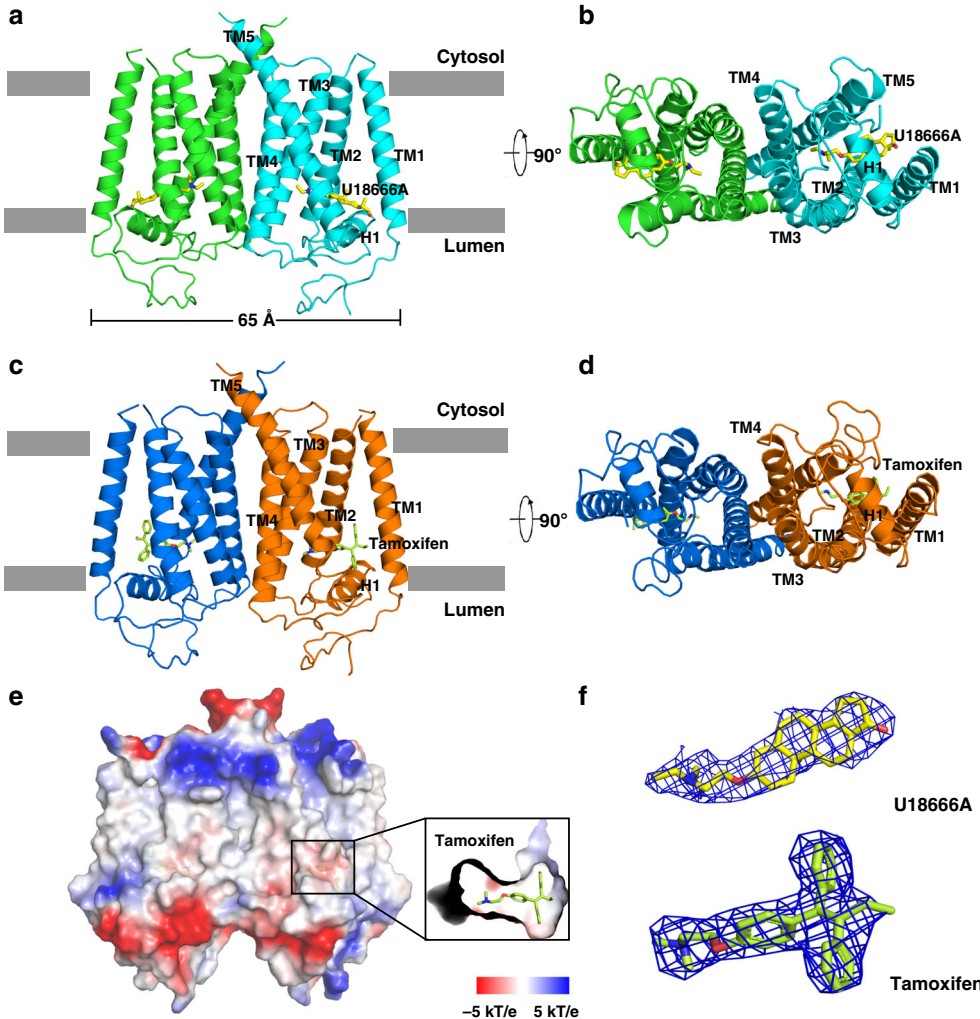

**Fig. 2** Molecular architecture of human EBP. **a** Overall structure of EBP with U18666A viewed parallel to the membrane. Each molecule of the dimer is indicated in the different colors. The two lines show the approximate location of the lipid bilayer. **b** The bottom view of EBP with U18666A from the lumen. **c** Overall structure of EBP with tamoxifen viewed parallel to the membrane. **d** The bottom view of EBP with tamoxifen from the lumen. **e** Electrostatic surface representation of EBP and the ligand-biding site with tamoxifen. **f**, *2Fo – Fc* map for U18666A and tamoxifen (blue mesh) contoured at 1σ

with our structural investigation (Supplementary Fig. 3). Crystals were only obtained with U18666A or tamoxifen in space group $P2_12_12$, owing to these two compounds presenting a high binding affinity to EBP (Supplementary Fig. 2). The structure was determined by selenium-based single-wavelength anomalous dispersion (SAD) and refined at 3.2 Å resolution with U18666A and 3.5 Å resolution with tamoxifen (Supplementary Figs. 4, 5 and Supplementary Table 1). This protein contains five transmembrane helices (TMs 1–5) (Fig. 2a–e). Based on the previous immunoblotting study[25] and positive inside rule[26], we assigned the N-terminal face to the lumen (Fig. 2a, b). In the density map, the ligand (U18666A or tamoxifen) was determined unambiguously (Fig. 2f). Comparison between U18666A-bound and tamoxifen-bound structures revealed a similar conformation (Supplementary Fig. 6).

A DALI search for structural homologues failed to identify a similar entry for the entire structure, implying that EBP presents an unreported fold. EBP forms a homodimer in the crystal, as reported in previous solution studies[27], with dimensions of 65 × 30 × 55 Å. The area of the dimer interface including TMs 3–5 of each monomer is 1460 Å[2] (Fig. 2a–d). To further validate the dimerization of EBP, we co-expressed His-EBP and Strep-EBP. Western blot showed that His-EBP can be pulled-down by Strep-

Tactin sepharose (Supplementary Fig. 7a), implying that EBP can form a dimer as shown by our structural observation. Mutations in the dimer interface lead to slower growth of the ΔErg2 *Saccharomyces cerevisiae* at exposure of 50 ng/ml cycloheximide (Supplementary Fig. 7b, c), implying that the dimerization may play a role in fully activating EBP. Notably, sequence analysis suggests that the TM6SF2 protein contains two repeat units, each of which shares sequence homology with EBP[21].

**The horizontal helix linker**. A linker between TM2 and TM3 forms a membrane horizontal helix (H1) that blocks a hydrophobic cavity created by the residues of TMs 2–5 (Fig. 2a–d). The residues of H1, which face the cavity, are hydrophobic and aromatic (Fig. 3a, b and Supplementary Fig. 8), while the residues facing the lumen are hydrophilic. Importantly, Ala substitutions of Trp101, Tyr104 and Tyr111, which face the hydrophobic cavity, abolish over 90% of the isomerase activity by gas chromatography analysis[28]. Conversion of Δ8-sterols to Δ[7]-isomers occurs through the uptake of solvent hydrogen from the membrane[29], implying that a proton or water molecule can enter the catalytic cavity with helix H1 possibly serving as the gate.

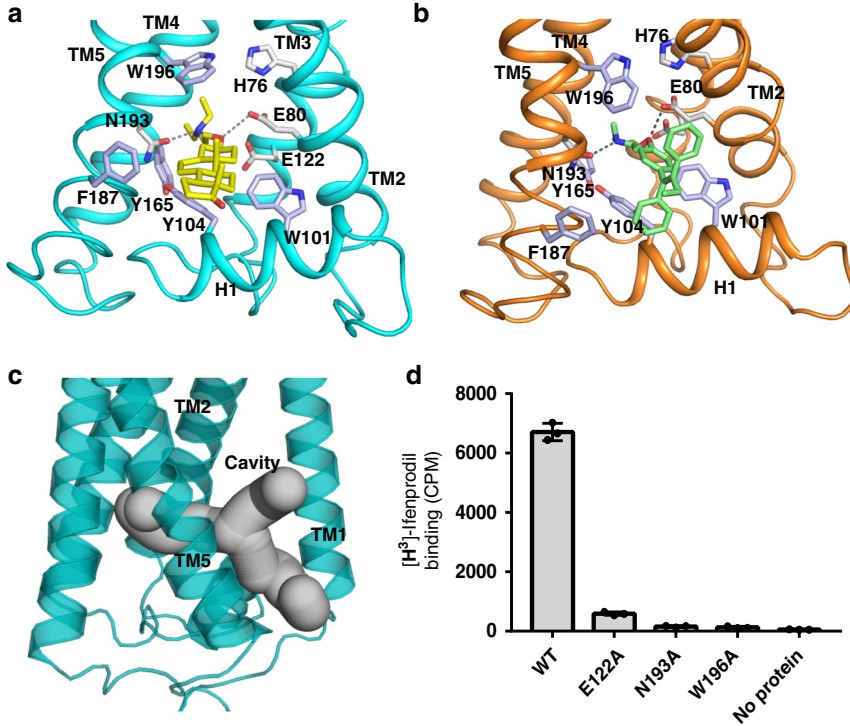

**Fig. 3** Molecular mechanism of compound binding to the membrane cavity. **a** Details of EBP binding to U18666A. The hydrophilic interactions are indicated by dashed lines. The residues related to substrate recognition are shown as sticks. **b** Details of EBP binding to tamoxifen, annotated as in **a**. **c** Overall view of the membrane cavity. **d** Binding of EBP mutants to [³H]-Ifenprodil. Data shown are the mean ± SD of triple times determinations. Source data are provided as a Source Data file

**The membrane cavity and the putative catalytic mechanism.** Structural analysis indicates that the cavity is created by the TMs that provide the hydrophobic environment to host the compound (Fig. 3c). There are four polar residues in this cavity suggesting a special role. Asn193 and Glu122 recognize the amine group of the ligand, which is stabilized through a π-cation interaction with Trp196 (Fig. 3a, b). Our binding assay also shows that mutation of Glu122, Asn193 or Trp196 decreases ligand-binding capacity (Fig. 3d). The 2.7 Å distance between the side chains of Glu80 and Glu122 and the 3.2 Å distance between the side chains of Glu80 and His76 allow a proton to transfer through the hydrogen-bonding network as a catalytic triad. As these three residues have a 3.5–5 Å distance to the compound, they are likely involved in the catalytic mechanism of EBP during cholesterol biosynthesis (Fig. 4a).

To further investigate the catalytic mechanism, we performed the molecular docking simulations with $\Delta^8$-sterol (Fig. 4a and Supplementary Fig. 9). These docking results along with previous studies[28,30] suggest a putative catalytic mechanism of EBP in cholesterol biosynthesis. Helix H1 can allow a proton or water molecule to access the cavity from the ER lumen. His76 may play the role of a proton donor residue, protonating $\Delta^8$-sterol at C9α with the subsequent generation of a carbonium ion at C8. The resulting carbocationic sterol intermediate would be stabilized by Trp196 through a π-cation interaction (Fig. 4a). Then, this intermediate would be deprotonated at C7β by Glu80 that would be stabilized by Glu122. After this reaction, the proton would be returned to His76 through the hydrogen-bonding network. The produced $\Delta^7$-isomers would be released into the membrane from the membrane gate between TM1 and TM5 (Fig. 3c).

This putative mechanism is analogous to the enzymatic action of ketosteroid isomerases, for which acidic residues act as a proton donor or acceptor[31], although the ketosteroid isomerase catalyzes the reaction in an aqueous environment. To validate our

hypothesis, we performed a functional analysis in yeast. Unlike wild-type EBP, mutants with His76Ala, Glu80Ala, Glu122Ala, or Trp196Ala did not survive at exposure of 50 ng/ml cycloheximide (Fig. 4b). We also purified EBP and its mutants from HEK293 cells, and the resulting protein was incubated with deuterium labeled zymostenol (zymostenol-d7). After the reaction, the mixture was analyzed by LC-MS/MS. The results show that mutants with His76Ala or Glu80Ala completely lose the enzymatic activity, while mutants with Glu122Ala or Trp196Ala retain a much lower activity than the wild-type protein (Fig. 4c and Supplementary Fig. 10). These results are consistent with the previous gas chromatography analysis, which demonstrated that EBP variants with any of these mutations essentially affected its isomerase activity[28].

## Discussion

Conradi–Hunermann syndrome is an inherited X-linked dominant variant of chondrodysplasia punctata, and is characterized by bone, skin, and eye abnormalities[8]. Biochemical studies on Conradi–Hunermann syndrome patients demonstrated increased amounts of 8,9-unsaturated sterols in the plasma and tissues due to mutations in the *EBP* gene[8,32]. Structural determination of EBP affords the opportunity to map these disease-related mutations[33] (Fig. 5 and Supplementary Table 2). Mutations, which are localized to the membrane cavity, may affect sterol binding/entry and catalytic reaction. Five mutations are found in H1 and the luminal domain, including L18P, E103K, A105D, G107E, R110Q. These mutations may block the solvent entry. Mutations in the dimerization interface may destabilize the protein or cause folding defect, subsequently affecting the activity of EBP. Therefore, our structures serve as a framework for future biophysical, biochemical, and cellular analysis of Conradi–Hunermann syndrome.

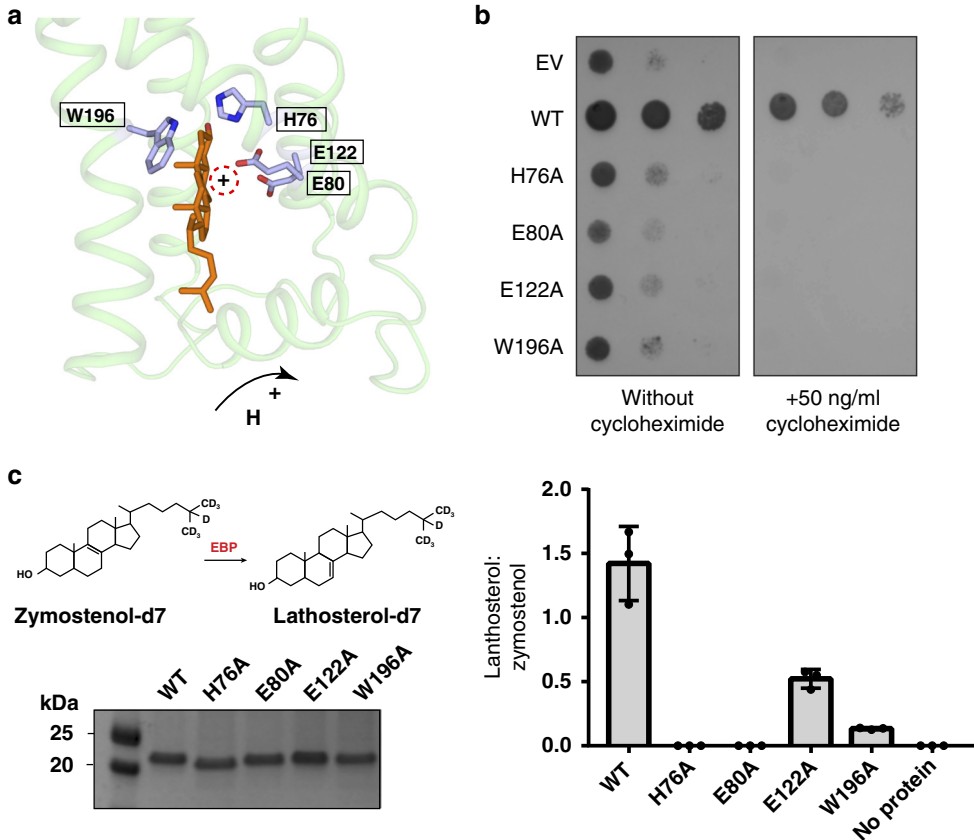

**Fig. 4** The putative catalytic mechanism of EBP in the isomerization. **a** A top-scoring model of EBP and $\Delta^8$-sterol that was generated by Molecular Docking Simulations. The key resides of isomerization by EBP are shown as sticks. **b** Yeast complementation assay. EBP with the catalytic mutation cannot rescue the growth of $\Delta$Erg2 *Saccharomyces cerevisiae*. Growth of yeast expressing EBP mutant in the presence of 50 ng/ml concentrations of cycloheximide for 24 to 48 h. **c** Enzymatic activity assay in vitro. EBP catalyzes the conversion of deuterium labeled zymostenol (zymostenol-d7) to lathosterol-d7. LC-MS/MS was used for isolation and analysis zymostenol-d7 and lathosterol-d7. The SDS-PAGE shows the purified recombinant proteins used in the assays. Data shown are the mean ± SD of triple times determinations. Source data are provided as a Source Data file

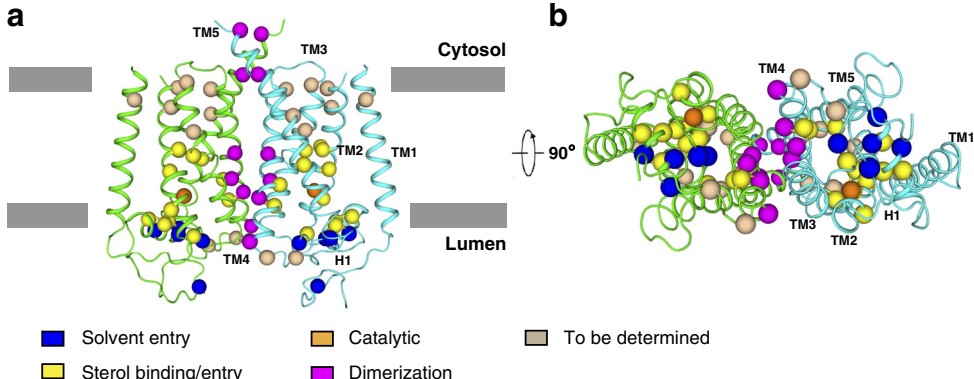

**Fig. 5** The locations of Conradi–Hunermann syndrome mutations. **a** Overall structure of EBP with disease-related mutations viewed parallel to the membrane. **b** The bottom view. The α carbon atoms of representative disease-related residues are shown as spheres. Color code for mutations: blue, solvent entry; yellow, sterol binding/entry; orange, catalytic sites; magenta, dimerization; wheat, structural consequences yet to be determined. The mutations are summarized in Supplementary Table 2

We also showed that U18666A and tamoxifen bind to EBP in the same manner, implying the binding mode of EBP ligands (Fig. 3a, b). The compounds that bind to EBP have a positively-charged amine group (Fig. 1c and Supplementary Fig. 2), mimicking the carbocationic reaction sterol intermediate (Fig. 4a). Trp196 is capable of stabilizing the compound through a π-cation interaction, while His76, Glu80, Glu122, and Asn193 may form electrostatic interactions and hydrogen bonds with the amine group, further enhancing the binding (Fig. 3a, b). Besides the amine group, the hydrophobicity of compounds enables them to partition into the membrane. The binding pocket in EBP consists of several conserved aromatic residues that provide a hydrophobic environment directly facing the membrane, allowing the cavity to host diversely structured ligands (Fig. 3c). Interestingly, some EBP ligands can bind to other enzymes in the cholesterol biosynthesis pathway. Those enzymes may use the same

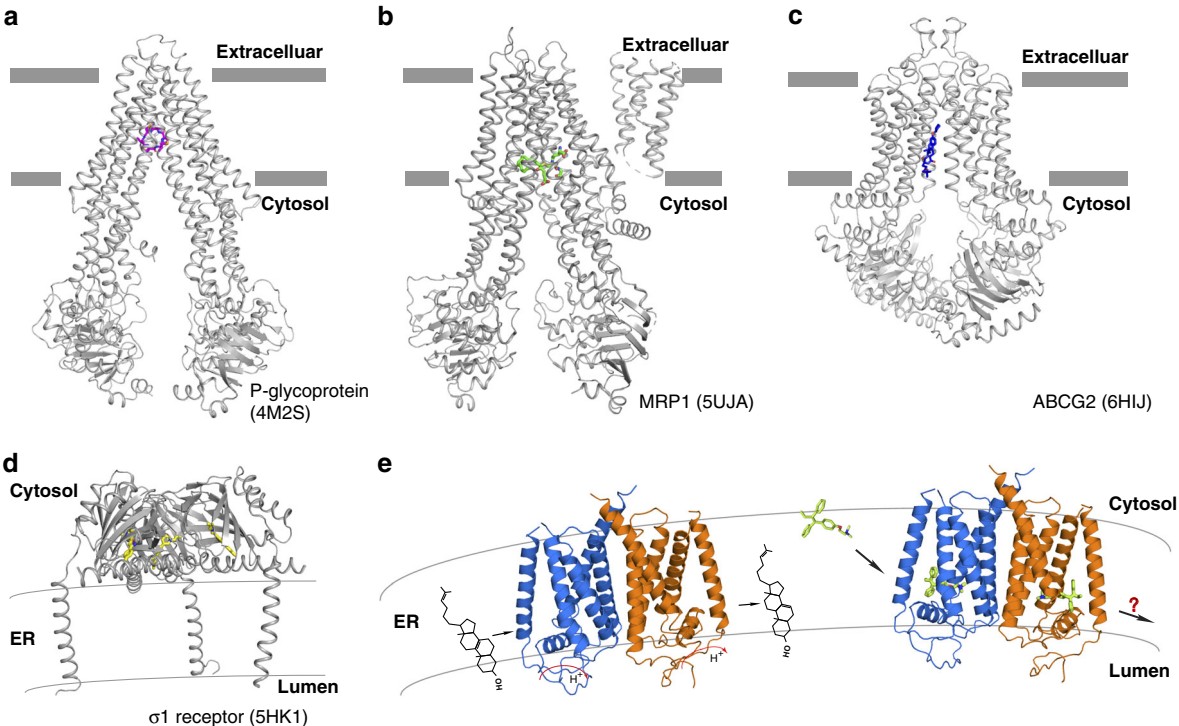

**Fig. 6** Structural comparison of EBP with other membrane proteins known to cause drug resistance. **a** Overall structure of P-glycoprotein (PDB code: 4M2S). The two lines show the approximate location of the lipid bilayer. The ligand is shown in stick representation. **b** Overall structure of MRP1 (PDB code: 5UJA). **c** Overall structure of ABCG2 (PDB code: 6HIJ). **d** Overall structure of σ1 receptor (PDB code: 5HK1). **e** The dual role of human EBP in the ER membrane

way as EBP to bind the inhibitors if they can generate the carbocationic sterol intermediate and have a large membrane cavity.

Several studies have indeed implied that EBP may serve as a multidrug resistance protein by decreasing the intracellular concentration of drugs including tamoxifen[16,34]. Comparison with ligand structures of other known multidrug binding proteins (P-glycoprotein (P-gp)[35], multidrug resistance-associated protein 1 (MRP1)[36], ABCG2[37] and σ1 receptor[38]) reveals that EBP shares a large cavity in the transmembrane domain for ligand-binding with P-gp, MRP1, and ABCG2 (Fig. 6a–c). By contrast, the σ1 receptor includes only one TM that recruits its extracellular domain to bind ligand (Fig. 6d). Similar to the σ1 receptor, the drug scavenger activity of EBP in interfering with drugs and the energy source are still unclear. It is clear that EBP in the ER is not able to directly export tamoxifen to the extracellular space. EBP might redistribute the drug to the cytosol, preventing its accumulation in the membrane, or release it into the ER lumen, followed by vesicle-mediated exocytosis to the extracellular space (Fig. 6e).

In summary, we report two crystal structures of human sterol isomerase with distinct pharmacologically active compounds. Blocking EBP activity has been shown to contribute to the death of tumor cells[9,10,17,18]. The structurally related σ2 receptor presents high expression levels in breast, colon and stomach cancers[20] and has been reported to be a potential therapeutic target for neurocognitive disorders[39]. This work provides the atomic picture that will allow us to target this enzyme and its homologues for further drug development and treatment of related human diseases.

## Methods

**Protein expression and purification**. The cDNA of human EBP (GI number 10682) was cloned into pEG BacMam with an N-terminal Strep-tag (for crystallization) or an N-terminal His-tag (for binding assays). DNA constructs were generated using two-step PCR. The protein was expressed using baculovirus-mediated transduction of mammalian HEK-293S GnTI⁻ cells (ATCC). At 48 h post infection, the cells were harvested by centrifugation and frozen at −80 °C until purification. For both tamoxifen and U18666A-bound proteins, 10 μM ligand was added in all purification steps. After thawing, cells were lysed by sonication in buffer A, containing 20 mM Hepes, pH 7.5, 150 mM NaCl with 1 mM PMSF and 5 μg/mL leupeptin. After low-speed centrifugation, the resulting supernatant was incubated in buffer A with 1% (w/v) Lauryl Matose Neopentyl Glycol (LMNG, Anatrace) for 1 h at 4 °C. The lysate was centrifuged again and the supernatant was loaded onto a Strep-Tactin affinity column (IBA) or a Ni²⁺-NTA affinity column (Qiagen). After washing with two column volumes, the protein was eluted in 20 mM Hepes, pH 7.5, 150 mM NaCl, 2.5 mM desthiobiotin or 300 mM imidazole, 0.01% LMNG, and concentrated. The concentrated protein was purified by Superdex 200 Increase size-exclusion chromatography column (GE Healthcare) in a buffer containing buffer A and 0.12% (w/v) Cymal-5 (Anatrace) or 0.01% LMNG. The peak fractions were collected and concentrated to 5~8 mg/ml for crystallization or binding assays.

Selenomethionine-labeled protein was expressed in HEK-293S GnTI⁻ cells as previously reported[40,41]. In brief, cells were pelleted and resuspended in DMEM without L-methionine, L-glutamine and L-cystine (Thermo Fisher scientific) supplemented with 10% FBS, 1 mM glutamate and 37.5 mg/L L-cystine-2HCl after 12 h infection. 60 mg of L-selenomethionine was added per liter of cells after depleting for 12 h. Cells were harvested 36 to 48 h after transduction and protein was purified using the same protocol as described above, except for the addition of 0.5 mM TCEP to all purification steps.

**Crystallization**. Crystals were grown at 20 °C by the hanging-drop vapor-diffusion method. Initially, the diffraction and quality of the crystals were poor. After optimization, crystals in $P2_12_12$ space group appeared within two days in a solution containing 0.1 M Hepes pH 7.5 or Tris pH 7.5, 20–23% (v/v) PEG 600. Each asymmetric unit contains one dimer and one monomer. Selenium derivatives were obtained in mother liquor with 2 mM TCEP. All crystals were flash-frozen in liquid nitrogen stream with 0.1 M Hepes pH 7.5 or Tris pH 7.5, 30% PEG 600 and 0.1 mM ligand for cryo-protection.

**Data collection and structure determination**. Diffraction data were collected at 100 K at NE-CAT and SBC-CAT of the Advanced Photon Source and FMX of the NSLS-II. All data sets were processed using HKL2000[42]. The anomalous signal in the Se-derivative data was further magnified with the local-scaling algorithm using the program SOLVE[43]. Then, the Se sites were determined using the program SHELXD[44]. The identified sites were refined and the initial phases were generated

in the program PHASER[45] with the SAD experimental phasing module. The initial model was manually built in COOT[46] manually. Large aromatic/hydrophobic residues were assigned initially to facilitate the register of the transmembrane helices. The U18666A-bound and tamoxifen-bound structures were refined with PHENIX.REFINE[47] at 3.2 and 3.5 Å resolution respectively. Due to the flexibility and limited resolution, the residues 1–6, 53–59 and 220–230 were not visible in the electron density map and were not included in the final structure. Model validation was performed with MolProbity[48]. The final models of U18666A-bound and tamoxifen-bound contained 93.07, 6.93 and 0.0%, and 91.09, 8.74, and 0.17% in the favored, allowed, and outlier regions of the Ramachandran plot, respectively. All figures were generated with PyMOL.

**Molecular docking simulations.** The Maestro platform was used to access modules of the Schrodinger software package for structure preparation and docking. The structural model of EBP was prepared using Protein Preparation Wizard and the PROPKA module was used to set the protonation state of the protein at pH 7.5. The center of the binding pocket was defined by residues that lie within a 10 Å radius of the cavity in the structure of EBP. 3D coordinates of zymersterol were generated with LigPrep using the EPIK module to set the pH to 7.5 and the OPLS_2005 force field option. The resulting ligand structure was then docked to the structural model of EBP using the Glide standard precision (SP) scoring function. The docking procedure yielded a single cluster of poses. The poses with the highest docking scores were chosen as representatives of the binding model.

**Yeast isomerase complementation assay.** Wild-type and mutant EBP were subcloned into the URA3 shuttle vector pCM190 (Euroscarf, Germany) (Supplementary Table 3). The plasmids were introduced in Erg2-deficient *Saccharomyces cerevisiae* strain Y17700 (Euroscarf) by electroporation. A single colony was picked from a URA⁻ selective plate. For the yeast rescue assay, the yeast was grown on URA⁻ plates in the absence or the presence of sub-inhibitory concentrations of cycloheximide (50 ng/ml) at 30 ℃ for 24 to 48 h. The yeast isomerase inhibition assay was followed the same protocol as above, except for the addition of 50 μM tamoxifen or 100 μM U18666A to the URA⁻ plates. The results were confirmed by three independent experiments with different colonies.

**SPA-based binding assay.** All SPA experiments were performed with Copper His-tag YSi Scintillation proximity beads (PerkinElmer, RPNQ0096). Beads were diluted to 2.5 mg/ml in 150 mM MES-NaOH, pH 6.5, 50 mM NaCl, 20% glycerol, 2 mM TCEP and 0.05% DDM. 100 nM [³H]-ifenprodil was incubated with 800 nM N-terminal His₆-tagged EBP protein in a total volume of 100 μl at 4 ℃ for 2 h. The solution was added to 100 μl SPA beads and incubated by vigorous shaking at 4 ℃ in the dark for 2 h. The mixture was loaded into individual wells of 96-well plates. Samples were measured in duplicate at room temperature with a Wallac 1450 MicroBeta plate PMT counter. For the competition assay, the concentration of tamoxifen or U18666A was increased from 0.1 nM to 1 mM. To define the non-specific binding activity, 400 mM imidazole was added to the wells to compete with the His₆-tag for bead binding. Nonspecific binding was subtracted from each data point. All experiments were performed at least three times and data are presented as mean ± s.d. Data fitting was performed using GRAPHPAD PRISM 5.0 Demo.

**Dimerization analysis.** Mammalian HEK-293S GnTI⁻ cells were co-infected with baculovirus containing N-terminal His-tag EBP gene and baculovirus containing N-terminal Strep-tag EBP gene. At 48 h post infection, the protein was purified with a Strep-Tactin affinity column (IBA) followed by gel filtration. To further verify the nonspecific binding between His-tag and Strep-Tactin, proteins were incubated at 4 ℃ with Strep-Tactin Sepharose for 2 h. After washing several times, samples were separated by SDS-PAGE and detected by Western blot using anti-His (Millipore, 05–949, 1:500) or anti-Strep antibody (Abcam, ab76950, 1:500).

**Enzymatic activity assay in vitro.** Standard assays for enzymatic activity of EBP and mutants were conducted as follows: all proteins were changed to buffer B (50 mM Tris/HCl, pH 7.5, 2 mM MgCl₂, 1 mM EDTA, 2 mM 2-mercaptoethanol, 5% glycerol and 0.1% TWEEN 80) by gel filtration. The assays were performed at 37 ℃ with gentle shaking for 12 h in 200 μl buffer B including 10 μM recombinant protein, 50 μM zymostenol-d7. The enzymatic reactions, quenched by 0.5 mL 6% (w/v) KOH-methanol, were brought to 2 mL using water/methanol 1:1. A surrogate standard of 600 ng d3-campesterol (30 p.p.m. in methanol) was added to monitor recovery. 1 mL of dichloromethane was added and the samples were vortexed and centrifuged at 2600 RCF for 5 min to induce phase separation. The organic phase was removed to a fresh tube, and a second 1 mL of dichloromethane was added to the aqueous phase. After a second vortexing and centrifugation, the organic phase was removed and pooled with the first extraction. The lipid extracts were dried under a flow of nitrogen with gentle heating and dissolved in 300 μL 90% methanol for LC-MS/MS. The substrate and product were analyzed as outlined in McDonald et al.[49]. using scheduled MRM transition of 376.4/161.2. Analytical standards (Avanti Polar Lipids) were used to confirm retention times of the analytes.

**Reporting Summary**. Further information on research design is available in the Nature Research Reporting Summary linked to this article.

## Data availability

Data supporting the findings of this manuscript are available from the corresponding author upon reasonable request. A reporting summary for this Article is available as a Supplementary Information file. The source data underlying Figs. 1d, 3d, 4c and Supplementary Fig. 7a are provided as a Source Data file. Atomic coordinates for the atomic model have been deposited in the Protein Data Bank under the accession numbers 6OHT (EBP-U188666A) and 6OHU (EBP-tamoxifen).

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

## Acknowledgements

This research was done in the memory of G. Blobel for his generosity and encouragement. We thank the NE-CAT and SBC-CAT at the Advanced Photon Source for assistance and FMX at the NSLS-II with data collection; L. Beatty for help with tissue culture and M. Brown, E. Debler, J. Goldstein and P. Schmiege for discussion. This work was supported by the Endowed Scholars Program in Medical Science of UT Southwestern Medical Center, O'Donnell Junior Faculty Funds, Welch Foundation (I-1957) and NIH grant P01 HL020948 (to J.M. and X.L.). X.L. is a Damon Runyon-Rachleff Innovator supported by the Damon Runyon Cancer Research Foundation (DRR-53–19) and a Rita C. and William P. Clements, Jr. Scholar in Biomedical Research at UT Southwestern Medical Center.

## Author contributions

X.L. conceived the project and designed the research with T.L. T.L. and A.H. performed the studies. B.T. and J.M. performed the mass spectrometry analysis. J.W. built the initial model. T.L. and X.L. analyzed the data and contributed to manuscript preparation. X.L. wrote the manuscript.

## Additional information

**Competing interests:** The authors declare no competing interests.

