## [Peer Review File · Nature Communications]

Reviewers' Comments:

Reviewer #1:

Remarks to the Author:

Li and coworkers report for the first time 2 crystal structures of human sterol 8 to 7 isomerase (also known as emopamil-binding protein-EBP), one bound with U18666A and another one bound with tamoxifen. Although these studies represent the second sterolic enzyme of cholesterol biosynthesis for which a three-dimensional structure has been determined- an important achievement in sterol biochemistry where X-ray structures of sterolic enzymes is lacking- the work is noticeably flawed, requiring more research to make the study suitable for publication in Nature Communications.

Major comments:

1) The paper is poorly written and relevant background literature is missing.

- Not cited is a recent review of cholesterol biosynthesis focusing on mechanistic enzymology: Nes, W. D. Biosynthesis of cholesterol and other sterols. *Chem. Rev.* 111, 6423 (2011).

2) There is no activity assay of pure enzyme showing the protein can convert native 8-ene substrate to 7-ene product (required to show recombinant enzyme acts in its physiological conformation such that the resulting crystal structure reflects its native dimer organization).

Additionally, the binding data of EBP and tamoxifen (potentially of a non-active enzyme) is not correlated to kinetic studies of isomerase inhibition by these drugs showing enzyme activity is affected.

-Not cited is a paper detailing for the first time, the cloning, functional expression and purification of human sterol isomerase. Nes, W. D. et al. Purification, characterization and catalytic properties of human sterol 8-isomerase. *Biochem. J.* 367, 587 (2002). This study has several features—the *k_{cat}* (turnover number) of the pure enzyme was determined, the reaction mechanism was determined unambiguously using 2H₂O and NMR analysis, the dissociations constants (*K_i*) for emopamil and tamoxifen (and several other drugs) determined and the equilibrium constant for the forward isomerization direction determined for the first time.

3) Crystal structures of related human enzyme, the sterol C14-demethylase (CYP51), bound with azoles drugs were not discussed. Additionally, refinement of crystal resolution of human sterol isomerase complexes with the two drugs is low at 3.5 angstroms while the corresponding density map at 1. σ unreliable.

Not cited is a paper detailing the crystal structure of human CYP51 bound with medical azoles. This paper could serve as a template for the authors on how to research and report a crystal structure of a sterolic enzyme bound with a drug. Hargrove, T. Y. et al. Human sterol 14 α -demethylase as a target for anticancer chemotherapy: towards structure-aided drug design. *J. Biol. Chem.* 57, 1552 (2016). Notably, in this paper the CYP51 structure was resolved to 2 angstroms, the authors performed binding studies of drugs and established the kinetic parameters of native substrate (*K_m/k_{cat}*) against the pure recombinant enzyme.

4) Limited mutagenesis study and no binding data for the native sterol substrate failed to support the speculative reaction mechanism already proven by Nes and others. Additionally, ligand binding assays to validate proposed isomerization activity of the human isomerase is not valid.

Not cited is a paper by the French group who in performing a systematic site-directed mutagenesis of the corn plant sterol 8-isomerase established residues in the active site associated with anchoring substrate and involved with the isomerization reaction likely to be the same for the human sterol isomerization reaction. Rahier, A. et al Identification of essential amino acid residues in a sterol 8-7 isomerase from *Zea mays* reveals functional homology and diversity with the isomerases of animal and fungi origin. *Biochem. J.* 414, 247 (2008). Importantly, these authors also developed an activity assay system using the recombinant corn 8—isomerase in yeast microsomes.

Minor comments:

1. Functional characterization approach used by the authors is not sufficient to support the recombinant enzyme is active in its native form. Several previous investigators have successfully

- purified the enzyme either using conventional purification protocols of the 1980's (Gaylor's group) or one involving pass through the erg2 knock out strain developed by Martin Bard (Nes' group).
2. The work would be substantively enhanced with the native substrate bound in the active site.
 3. Limited mutagenesis with no corresponding activity assay.
 4. Rationale and experimental data unclear for Figures 4 through 6.

Reviewer #2:

Remarks to the Author:

Long, Wang and Li report the first structure of the human sterol isomerase, which catalyses one of a dozen or more cholesterol biosynthesis steps post-lanosterol. This sterol isomerase is of clinical significance: defective in an inborn error of metabolism (Conradi-Hunerman syndrome), and binds with high affinity to a diverse array of drugs and hence is also known as emopamil-binding protein (EBP). The authors report two crystal structures of human EBP, one with tamoxifen (anti-breast cancer drug) and the other with U18666A (cholesterol synthesis inhibitor), revealing five transmembrane domains that forms a cavity in the membrane to accommodate diverse ligands.

Very few structures are available for cholesterol synthesis enzymes and so this work will be of considerable interest to others in the field and beyond. However, as it currently stands the manuscript falls short in a number of key areas.

1. Did the authors attempt to obtain structures with the actual substrates (zymosterol/zymostenol)? Obviously, this would be more useful in being able to reveal the mechanism of the sterol isomerisation than structures with drugs. If no such structure(s) was/is forthcoming, the work would be strengthened by performance of molecular dynamic simulations with at least one of the substrates.
2. Why were tamoxifen and U18666A chosen over emopamil which would seem more logical (since the enzyme is best known as EBP)? This needs more justification.
3. How is the ligand binding cavity able to bind such diverse structures as shown in Supp Fig. 2? Many have bulky aromatic groups on either side of the amine, rather than just on one end as in tamoxifen and U18666A. Apart from them being tertiary amines, hydrophobicity must be another determinant given that these compounds should presumably partition into the membrane. This information could be added to Supp Fig. 2 and should be discussed, also in context of the apparently promiscuous ligand binding cavity.
4. How do the authors know the orientation of the protein in the membrane? Fig. 2 appears to show that the N-terminal is luminal whereas the C-terminus is cytosolic. However, it is unclear if this is based on any experimental evidence. If not, protease protection assays should be performed to substantiate this orientation.
5. Similarly, do the authors have data confirming the much earlier work (JBC 1994) that EBP indeed occurs as a homodimer?
6. The authors should briefly mention the genetics of Conradi-Hunerman syndrome (AKA X-linked dominant chondrodysplasia punctate). The reference used for the EBP mutants shown in Fig 5 seems is 7 years old (2012). The authors should update these mutations based on appropriate databases so as to ensure all known mutants are incorporated. It would be useful to include a Supp Table detailing the mutants with frequency, effect on activity/likely consequence, reference etc. It would also make more of a rounded story if select mutants (perhaps most reported) are tested in some way in terms of their functional consequences informed by the structure. E.g. an inability to dimerise.
7. Some of the sterol synthesis inhibitors that bind EBP also inhibit other steps of the pathway (e.g. U18666A inhibits lanosterol synthase, triparanol inhibits DHCR24). How does this fit in with the author's proposed mechanism of action for EBP? Do they propose that it occurs via a similar mode?
8. The speculation about EBP exporting drugs from the ER to the cytosol could be readily tested.
9. Can the authors differentiate between a proton vs water molecule in their proposed mechanism of action?

10. Some of the English expression can be tidied up throughout e.g. "Cholesterol... aids in the biosynthesis of...") Of course, it is the precursor for these compounds (though not of Vitamin D).

Response to Referees:

Reviewer #1

Li and coworkers report for the first time 2 crystal structures of human sterol 8 to 7 isomerase (also known as emopamil-binding protein-EBP), one bound with U18666A and another one bound with tamoxifen. Although these studies represent the second sterolic enzyme of cholesterol biosynthesis for which a three-dimensional structure has been determined- an important achievement in sterol biochemistry where X-ray structures of sterolic enzymes is lacking- the work is noticeably flawed, requiring more research to make the study suitable for publication in Nature Communications.

Major comments:

1) The paper is poorly written and relevant background literature is missing.

- Not cited is a recent review of cholesterol biosynthesis focusing on mechanistic enzymology: Nes, W. D. Biosynthesis of cholesterol and other sterols. Chem. Rev. 111, 6423 (2011).

Response: Point accepted. We apologize that we have missed several important pieces of literature in the field and we have cited this literature in the revision (ref. 6, line 37).

2) There is no activity assay of pure enzyme showing the protein can convert native 8-ene substrate to 7-ene product (required to show recombinant enzyme acts in its physiological conformation such that the resulting crystal structure reflects its native dimer organization). Additionally, the binding data of EBP and tamoxifen (potentially of a non-active enzyme) is not correlated to kinetic studies of isomerase inhibition by these drugs showing enzyme activity is affected.

-Not cited is a paper detailing for the first time, the cloning, functional expression and purification of human sterol isomerase. Nes, W. D. et al. Purification, characterization and catalytic properties of human sterol 8-isomerase. Biochem. J. 367, 587 (2002). This study has several features—the kcat (turnover number) of the pure enzyme was determined, the reaction mechanism was determined unambiguously using 2H2O and NMR analysis, the dissociations constants (Ki) for emopamil and tamoxifen (and several other drugs) determined and the equilibrium constant for the forward isomerization direction determined for the first time.

Response: The major finding of our manuscript is the reporting of the EBP structures. We purified human EBP directly from human cells, not yeast or bacterial cells which previous groups have done. Therefore, our protein more likely represents its physiological conformation. The isomerase assay, which we used in the current manuscript, has been established by Silve et al. J. Bio. Chem 37:22434 and Ashmana et al. Lipids 26:628. The mutagenesis studies on the active site of EBP can abolish the enzymatic activity in yeast (Fig. 4b) and these results are consistent with the previous gas chromatography analysis (Moebius et al, Biochemistry 38, 1119); therefore, we believe that our structure represents the real enzyme state in cells. For the binding data of EBP and the compounds, we would like to show that EBP could bind to the compounds *in vitro* as seen in previous studies (Fig. 1d). The yeast isomerase assay (Fig. 1b) shows that the compounds can prevent the growth of the yeast by inhibiting the isomerase activity. Combined with our structural observation, we conclude that these compounds may bind the catalytic site of EBP to block its isomerase activity. We apologize that we did not cite this

important paper (Nes et al. Biochem. J. 367:587) and we have cited it in our revision (ref. 14, line 48).

3) Crystal structures of related human enzyme, the sterol C14-demethylase (CYP51), bound with azoles drugs were not discussed. Additionally, refinement of crystal resolution of human sterol isomerase complexes with the two drugs is low at 3.5 angstroms while the corresponding density map at 1. σ unreliable.

Not cited is a paper detailing the crystal structure of human CYP51 bound with medical azoles. This paper could serve as a template for the authors on how to research and report a crystal structure of a sterolic enzyme bound with a drug. Hargrove, T. Y. et al. Human sterol 14 α -demethylase as a target for anticancer chemotherapy: towards structure-aided drug design. J. Biol. Chem. 57, 1552 (2016). Notably, in this paper the CYP51 structure was resolved to 2 angstroms, the authors performed binding studies of drugs and established the kinetic parameters of native substrate (K_m/k_{cat}) against the pure recombinant enzyme.

Response: We have two structures reported in our current manuscript. The resolution of EBP with U18666A is 3.2 angstroms and the resolution of EBP with tamoxifen is 3.5 angstroms. We screened over 500 crystals to obtain this data and we believe that we can not push the resolution anymore at this stage. We hope that this referee could realize the technical difficulty for the crystallization of integral membrane proteins and we also would like to indicate here, the R_{free} and Ramachandran plot of our two structures are reasonable and suitable for publication. CYP51 is a signal transmembrane protein and the ligand-binding pocket is located in the soluble domain. In contrast, EBP contains a larger transmembrane domain and it employs the transmembrane region to accommodate its ligand introducing more technical challenges for us to improve the resolution. We would like to point out the “J. Biol. Chem. 57, 1552 (2016)” in the comments should be “J. Lipid. Res. 57, 1552”. We have cited this work (ref. 7, line 38) in our revision.

4) Limited mutagenesis study and no binding data for the native sterol substrate failed to support the speculative reaction mechanism already proven by Nes and others. Additionally, ligand binding assays to validate proposed isomerization activity of the human isomerase is not valid. Not cited is a paper by the French group who in performing a systematic site-directed mutagenesis of the corn plant sterol 8-isomerase established residues in the active site associated with anchoring substrate and involved with the isomerization reaction likely to be the same for the human sterol isomerization reaction. Rahier, A. et al Identification of essential amino acid residues in a sterol 8-7 isomerase from Zea mays reveals functional homology and diversity with the isomerases of animal and fungi origin. Biochem. J. 414, 247 (2008). Importantly, these authors also developed an activity assay system using the recombinant corn 8-isomerase in yeast microsomes.

Response: Moebius et al. (Biochemistry 38:1119) established the mass spectrum analysis to screen residues of EBP, which are essential for triggering the isomerase reaction. Fortunately, our structural observation supports both theirs and the above French group’s findings. We also have performed the binding assay (Fig. 3d) and yeast isomerase assay (Fig. 4b) to validate these key residues according to our structural observations. We have cited Biochem. J. 414, 247 (ref. 30, line 129) in the revision.

For the binding to native sterol substrate, we have performed the molecular docking simulations with zymosterol (Fig. 4a and Supplementary Fig. 9) as the previous study (Shreya et al Cell 176:1040). This structural model of EBP bound to zymosterol gives support to our putative catalytic mechanism. The orientation of zymosterol allow His76 to protonate at C9 α with the subsequent generation of the carbocationic sterol intermediate, which can be stabilized by Trp196. Then, this intermediate would be deprotonated by Glu80. After this reaction, the proton would be recycled through the hydrogen-bonding network (Fig. 4a). We have discussed this point in the revision (lines 127-138).

Minor comments:

1. Functional characterization approach used by the authors is not sufficient to support the recombinant enzyme is active in its native form. Several previous investigators have successfully purified the enzyme either using conventional purification protocols of the 1980's (Gaylor's group) or one involving pass through the erg2 knock out strain developed by Martin Bard (Nes' group).

Response: Please see our response to the major point #2.

2. The work would be substantively enhanced with the native substrate bound in the active site.

Response: Point accepted. Please see our response to the major point #4.

3. Limited mutagenesis with no corresponding activity assay.

Response: Please see our response to the major point #4.

4. Rationale and experimental data unclear for Figures 4 through 6.

Response: Point accepted. We have provided the more details of our experiments in the methods and supplementary information to clarify these figures.

The authors thank this referee for his/her time and constructive comments.

Reviewer #2:

Long, Wang and Li report the first structure of the human sterol isomerase, which catalyses one of a dozen or more cholesterol biosynthesis steps post-lanosterol. This sterol isomerase is of clinical significance: defective in an inborn error of metabolism (Conradi-Hunerman syndrome), and binds with high affinity to a diverse array of drugs and hence is also known as emopamil-binding protein (EBP). The authors report two crystal structures of human EBP, one with tamoxifen (anti-breast cancer drug) and the other with U18666A (cholesterol synthesis inhibitor), revealing five transmembrane domains that forms a cavity in the membrane to accommodate diverse ligands.

Very few structures are available for cholesterol synthesis enzymes and so this work will be of considerable interest to others in the field and beyond. However, as it currently stands the manuscript falls short in a number of key areas.

1. Did the authors attempt to obtain structures with the actual substrates (zymosterol/zymostenol)? Obviously, this would be more useful in being able to reveal the mechanism of the sterol isomerisation than structures with drugs. If no such structure(s) was/is forthcoming, the work would be strengthened by performance of molecular dynamic simulations with at least one of the substrates.

Response: Point accepted. The substrate binding is stabilized by the transient positive charge of the intermediate sterol substrate; therefore, it is almost impossible for us to capture this state by X-ray crystallography. We have performed the molecular docking simulations with zymosterol (Fig. 4a and Supplementary Fig. 9) as the previous study (Shreya et al Cell 176:1040). This structural model of EBP bound to zymosterol gives support to our putative catalytic mechanism. The orientation of zymosterol allow His76 to protonate at C9 α with the subsequent generation of the carbocationic sterol intermediate, which can be stabilized by Trp196. Then, this intermediate would be deprotonated by Glu80. After this reaction, the proton would be recycled through the hydrogen-bonding network (Fig. 4a). We have discussed this point in the revision (lines 127-138).

2. Why were tamoxifen and U18666A chosen over emopamil which would seem more logical (since the enzyme is best known as EBP)? This needs more justification.

Response: We tried to grow the crystals of EBP with different ligands including AY-9944 and Trifluoperazine; however, we were only able to obtain good crystals with tamoxifen and U18666A. This may be because EBP has a higher binding affinity to tamoxifen and U18666A than to these compounds as show in Supplementary Fig. 2. Therefore, we didn't choose emopamil for structural observation as its binding affinity to EBP is not higher than that of tamoxifen or U18666A (Supplementary Fig. 2). We have pointed it out in lines 80-82.

3. How is the ligand binding cavity able to bind such diverse structures as shown in Supp Fig. 2? Many have bulky aromatic groups on either side of the amine, rather than just on one end as in tamoxifen and U18666A. Apart from them being tertiary amines, hydrophobicity must be another determinant given that these compounds should presumably partition into the membrane. This

information could be added to Supp Fig. 2 and should be discussed, also in context of the apparently promiscuous ligand binding cavity.

Response: Point accepted. As shown in Fig. 3c, the cavity is big enough to bind diversely structured ligands. The amine group of the ligands is the primary determinant to bind to the cavity, while hydrophobicity of ligands is another determinant. We would like to emphasize here that the amine group of the various compounds mimics the intermediate state of the sterol substrate (Fig. 4a); this feature causes EBP to recognize these different ligands. The binding pocket in EBP consists of several conserved aromatic residues that provide a hydrophobic environment directly facing the membrane, allowing the cavity to host diversely structured ligands (Fig. 3c). We have discussed this point in the revision (lines 169-172).

4. How do the authors know the orientation of the protein in the membrane? Fig. 2 appears to show that the N-terminal is luminal whereas the C-terminus is cytosolic. However, it is unclear if this is based on any experimental evidence. If not, protease protection assays should be performed to substantiate this orientation.

Response: Point accepted. Dussossoy D. et al. (Eur. J. Biochem. 263: 377) confirmed the orientation of EBP in the membrane by immunoblotting with a specific antibody targeting the N-terminal domain of EBP. We also use the “positive inside rule” (Nature 341: 456) to confirm this observation. We have cited these two pieces of literature (refs. 25 and 26, line 87) in the revision to verify this issue.

5. Similarly, do the authors have data confirming the much earlier work (JBC 1994) that EBP indeed occurs as a homodimer?

Response: Point accepted. To verify the dimer state, we have performed a pull-down assay to show that His-tagged EBP and Strep-tagged EBP can form dimer (see Supplementary Fig. 7a) and we have introduced mutations onto the dimer interface and performed the yeast assay to verify the physiological importance of the EBP dimer *in vivo* (see Supplementary Fig. 7c).

6. The authors should briefly mention the genetics of Conradi-Hunerman syndrome (AKA X-linked dominant chondrodysplasia punctata). The reference used for the EBP mutants shown in Fig 5 seems is 7 years old (2012). The authors should update these mutations based on appropriate databases so as to ensure all known mutants are incorporated. It would be useful to include a Supp Table detailing the mutants with frequency, effect on activity/likely consequence, reference etc. It would also make more of a rounded story if select mutants (perhaps most reported) are tested in some way in terms of their functional consequences informed by the structure. E.g. an inability to dimerise.

Response: Point accepted. We have pointed out the genetics of Conradi-Hunerman syndrome in PAGE 8. As this reviewer’s suggestion, we have updated the disease-related mutations of EBP based on the “ClinVar” database and recent studies. We have incorporated all known mutants in Fig. 5 and Supplementary Table 2.

7. Some of the sterol synthesis inhibitors that bind EBP also inhibit other steps of the pathway

(e.g. U18666A inhibits lanosterol synthase, triparanol inhibits DHCR24). How does this fit in with the author's proposed mechanism of action for EBP? Do they propose that it occurs via a similar mode?

Response: It is interesting that EBP ligands can also inhibit other steps of the cholesterol biosynthesis pathway. These inhibitors have a positively-charged amine group, mimicking the carbocationic reaction sterol intermediate of EBP, and the membrane cavity of EBP is big enough to host diversely structured inhibitors. Therefore, the other enzymes in the cholesterol biosynthesis pathway may use another mode to bind to the inhibitors if they can't generate the carbocationic sterol intermediate or have a small membrane cavity.

8. The speculation about EBP exporting drugs from the ER to the cytosol could be readily tested.

Response: The drug scavenger activity of EBP in drug interference and the energy source remain unknown. After checking the literature, there is no efficient approach to detect the drug efflux by EBP. We have pointed this out and emphasized that the mechanism in Fig. 6d is a "putative" mechanism.

9. Can the authors differentiate between a proton vs water molecule in their proposed mechanism of action?

Response: Wilton et al. (Biochem. J. 114: 71) and Lee et al. (J. Biol. Chem. 244: 2033) showed that the conversion of Δ^8 -cholestenol to Δ^7 -cholestenol occurs through the uptake of solvent hydrogen from the medium. However, we haven't had an ideal assay to differentiate between a proton and water molecule yet.

10. Some of the English expression can be tidied up throughout e.g. "Cholesterol... aids in the biosynthesis of...") Of course, it is the precursor for these compounds (though not of Vitamin D).

Response: Point accepted. We have polished our expression in the revision.

The authors thank this referee for his/her time and constructive comments.

Reviewers' Comments:

Reviewer #1:

Remarks to the Author:

In this revised submission to Nature Communications paper titled "Structural basis for human sterol isomerase in cholesterol biosynthesis and multidrug design", Li and coworkers report a vastly improved presentation of the background literature and adequately address some concerns discussed by the reviewers. The work described here, for the first time, details the three-dimensional structure of human sterol isomerase. The study is novel and potentially has broad impacts. However, the authors' unwillingness to perform a few more experiments to justify speculations on the physiological conformation of cloned protein, sterol binding and the isomerization reaction vagaries performed by this medically-relevant catalyst dampens enthusiasm of this reviewer to support publication. My concerns are as follows:

1) Since the authors have a purification procedure to generate homogenous human sterol isomerase, then it is their responsibility to show it is active, thereby confirming the X-ray structure is based on its physiological conformation. Nes' activity assay published in the Biochemical Journal paper is straightforward. All the authors need to do is assay zymosterol (commercially available) with the pure protein and analyze the enzyme-generated product by GC-MS. They should show chromatogram (establishes yield of conversion) and mass spectrum (established identity) of the product. UTSW has the sterol chemistry faculty to assist in this experiment.

2) The equivalence of the three-dimensional structure of human and yeast isomerase representing animal and fungal type enzyme is conjecture, particularly since this class of sterol enzyme has no structure for reference. Therefore, some doubt is raised when comparing the yeast mutation studies to the human sterol isomerase active site. Preparation of the relevant mutants discussed in the paper and analysis of the pure recombinant protein should also be straightforward. Binding studies of zymosterol against these mutants would support the speculations put forward in the Discussion.

3) This reviewer is well-aware of the struggles to generate membrane proteins for X-ray structure and is sympathetic on this matter. Yet, the limited value of a 3.2 to 3.5 angstrom structure is strengthened by the inability of the authors to accurately define the topography of the active site residues in relation to the sterol substrate, which curiously is not part of the study. To discuss mechanism warrants the native substrate complexed with the enzyme. Since the authors are adamant about discussing the sterol isomerization reaction then they should understand the reaction and the different isomers variably generated in nature from the isomerization reaction. Not only can the 8 move to 7, but both 8 and 7 can rearrange into the 8(14)-position (the thermodynamically favored isomer); bacteria operate the 8(9)- to 8(14)-isomerization while nematodes operate the 7 to 8(14)-isomerization. In your mechanistic analysis, you fail to consider what residues might be involved in the back reaction (demonstrated by Nes) and whether the human isomerase has the capacity to generate alternate isomers. A more thorough structure-based examination of the conformation and specific residues in your "catalytic triad" of the active site controlling the specificity of the reaction in humans would be useful for an exacting mechanistic analysis. The relation of keto-enol tautomerism to sterol isomerization is a new one for me! For the point of water versus proton in the reaction mechanism queried by Reviewer 2, I encourage you to read the Nes paper in Biochemical Journal where they reported the deuterium from 2H₂O is introduced at C9. Their work is definitive using high field NMR to prove the structure of the enzyme-generated 2H-product. Could you study 2H₂O against zymosterol with the pure isomerase, then analyze by X-ray?

Reviewer #2:

Remarks to the Author:

I'm satisfied with the majority of the authors' responses to my comments.

I encourage them to incorporate their response to my Point 7 briefly in the text, as other readers may well have the same question.

7. Some of the sterol synthesis inhibitors that bind EBP also inhibit other steps of the pathway (e.g. U18666A inhibits lanosterol synthase, triparanol inhibits DHCR24). How does this fit in with the author's proposed mechanism of action for EBP? Do they propose that it occurs via a similar mode?

Line 31: As indicated in my previous report please rephrase to:

Cholesterol maintains membrane structure, is a precursor in the biosynthesis of steroid hormones and bile acids...

Response to Referees:

Reviewer #1:

In this revised submission to Nature Communications paper titled "Structural basis for human sterol isomerase in cholesterol biosynthesis and multidrug design", Li and coworkers report a vastly improved presentation of the background literature and adequately address some concerns discussed by the reviewers. The work described here, for the first time, details the three-dimensional structure of human sterol isomerase. The study is novel and potentially has broad impacts. However, the authors' unwillingness to perform a few more experiments to justify speculations on the physiological conformation of cloned protein, sterol binding and the isomerization reaction vagaries performed by this medically-relevant catalyst dampens enthusiasm of this reviewer to support publication. My concerns are as follows:

1) Since the authors have a purification procedure to generate homogenous human sterol isomerase, then it is their responsibility to show it is active, thereby confirming the X-ray structure is based on its physiological conformation. Nes' activity assay published in the Biochemical Journal paper is straightforward. All the authors need to do is assay zymosterol (commercially available) with the pure protein and analyze the enzyme-generated product by GC-MS. They should show chromatogram (establishes yield of conversion) and mass spectrum (established identity) of the product. UTSW has the sterol chemistry faculty to assist in this experiment.

Response: Point accepted. We purified the human EBP from HEK293 cells, and the resulting proteins were incubated with the deuterium labeled zymostenol at 37 degrees. After the reaction, the mixture was extracted by organic solvent. We analyzed the lipid extracts by LC-MS/MS. The result shows that the wild type protein can convert over 60% of the precursor presenting the isomerase activity in a detergent solution (Fig. 4c and Supplementary Fig. 10). It concludes that our structure presents the real state of this enzyme in cells.

2) The equivalence of the three-dimensional structure of human and yeast isomerase representing animal and fungal type enzyme is conjecture, particularly since this class of sterol enzyme has no structure for reference. Therefore, some doubt is raised when comparing the yeast mutation studies to the human sterol isomerase active site. Preparation of the relevant mutants discussed in the paper and analysis of the pure recombinant protein should also be straightforward. Binding studies of zymosterol against these mutants would support the speculations put forward in the Discussion.

Response: Point accepted. We have generated the human EBP variants and tested their isomerase activities according to the approach above (please see our response to Point 1). The results show that the H76A and E80A mutants completely lose the enzymatic activity; E122A and W196A mutants only retain 30% and 10% activity of the wild-type protein, respectively (Fig. 4c and Supplementary Fig. 10). This is also consistent with the result of the yeast assay (Fig. 4b). The LC-MS/MS results have provided direct evidence of how EBP mutants bind the substrate to trigger the reactions; therefore, we have placed the result in the main text (Lines 146-153).

3) This reviewer is well-aware of the struggles to generate membrane proteins for X-ray structure and is sympathetic on this matter. Yet, the limited value of a 3.2 to 3.5 angstrom structure is strengthened by the inability of the authors to accurately define the topography of the active site residues in relation to the sterol substrate, which curiously is not part of the study. To discuss mechanism warrants the native substrate complexed with the enzyme. Since the authors are adamant about discussing the sterol isomerization reaction then they should understand the reaction and the different isomers variably generated in nature from the isomerization reaction. Not only can the 8 move to 7, but both 8 and 7 can rearrange into the 8(14)-position (the thermodynamically favored isomer); bacteria operate the 8(9)- to 8(14)-isomerization while nematodes operate the 7 to 8(14)-isomerization. In your mechanistic analysis, you fail to consider what residues might be involved in the back reaction (demonstrated by Nes) and whether the human isomerase has the capacity to generate alternate isomers. A more thorough structure-based examination of the conformation and specific residues in your "catalytic triad" of the active site controlling the specificity of the reaction in humans would be useful for an exacting mechanistic analysis. The relation of keto-enol tautomerism to sterol isomerization is a new one for me!

Response: The substrate binding of EBP is stabilized by the transient positive charge of the intermediate sterol substrate; therefore, it is almost impossible for us to capture this state by X-ray crystallography. We have performed the molecular docking simulations with the substrate. To validate the putative residues involved in the reaction, we performed the yeast assay and the LC-MS/MS analysis of the isomerization by EBP. The results support our hypothesis and are consistent with the previous studies on EBP (Biochemistry 38:1119; Biochem. J. 414:247). Nes et al (Biochem. J. 367:587, Fig. 5) had shown that the efficiency of the forward reaction is 10-fold higher than that of the back reaction. Although EBP catalyzes 7 and 8 back reaction and may have the capacity to generate alternate isomers, the major role of EBP is to catalyze 8 and 7 forward reaction under physiological conditions. Our main finding is to report the structures of EBP with distinct compounds revealing the characterization of the EBP transmembrane pocket for the native substrate and pharmacological compounds. The possibility that EBP can generate alternate isomers as well as the mechanism of the back reaction require further investigation.

For the point of water versus proton in the reaction mechanism queried by Reviewer 2, I encourage you to read the Nes paper in Biochemical Journal where they reported the deuterium from $^2\text{H}_2\text{O}$ is introduced at C9. Their work is definitive using high field NMR to prove the structure of the enzyme-generated 2H-product. Could you study $^2\text{H}_2\text{O}$ against zymosterol with the pure isomerase, then analyze by X-ray?

Response: We thank the suggestion of the referee on this point. The study on $^2\text{H}_2\text{O}$ against zymosterol with the pure isomerase by X-ray is not technically possible, since the substrate binding of EBP is stabilized by the transient positive charge of the intermediate sterol substrate. It is almost impossible to capture this state by X-ray crystallography.

The authors thank this referee for his/her time and constructive comments.

Reviewer #2:

I'm satisfied with the majority of the authors' responses to my comments.

I encourage them to incorporate their response to my Point 7 briefly in the text, as other readers may well have the same question 7. Some of the sterol synthesis inhibitors that bind EBP also inhibit other steps of the pathway (e.g. U18666A inhibits lanosterol synthase, triparanol inhibits DHCR24). How does this fit in with the author's proposed mechanism of action for EBP? Do they propose that it occurs via a similar mode?

Response: Point accepted. We have placed this response in the revision (Lines 179-182).

Line 31: As indicated in my previous report please rephrase to:

Cholesterol maintains membrane structure, is a precursor in the biosynthesis of steroid hormones and bile acids...

Response: Point accepted. We have rephrased the sentence as this referee's suggestion.

The authors thank this referee for his/her time and constructive comments.

Reviewers' Comments:

Reviewer #1:

Remarks to the Author:

The authors are to be congratulated on revising the paper in such a way to make the work experimentally sound. In its current form, the paper is excellent.